# ORION software tool for the geometrical calibration of all-sky cameras

**Juan Carlos Antuña-Sánchez**[1]*, **Roberto Román**[1], **Juan Luis Bosch**[2],
**Carlos Toledano**[1], **David Mateos**[1], **Ramiro González**[1], **Victoria Cachorro**[1], **Ángel de Frutos**[1]

**1** Group of Atmospheric Optics, Universidad de Valladolid (GOA-UVa), Valladolid, Spain, **2** Departamento de Química y Física, Universidad de Almería, Almería, Spain

* jcantuna@goa.uva.es

## Abstract

This paper presents the software application ORION (All-sky camera geOmetry calibRation from star positIONs). This software has been developed with the aim of providing geometrical calibration to all-sky cameras, i.e. assess which sky coordinates (zenith and azimuth angles) correspond to each camera pixel. It is useful to locate bodies over the celestial vault, like stars and planets, in the camera images. The user needs to feed ORION with a set of cloud-free sky images captured at night-time for obtaining the calibration matrices. ORION searches the position of various stars in the sky images. This search can be automatic or manual. The sky coordinates of the stars and the corresponding pixel positions in the camera images are used together to determine the calibration matrices. The calibration is based on three parameters: the pixel position of the sky zenith in the image; the shift angle of the azimuth viewed by the camera with respect to the real North; and the relationship between the sky zenith angle and the pixel radial distance regards to the sky zenith in the image. In addition, ORION includes other features to facilitate its use, such as the check of the accuracy of the calibration. An example of ORION application is shown, obtaining the calibration matrices for a set of images and studying the accuracy of the calibration to predict a star position. Accuracy is about 9.0 arcmin for the analyzed example using a camera with average resolution of 5.4 arcmin/pixel (about 1.7 pixels).

## 1 Introduction

All-sky cameras are ground-based instruments capable of capturing images of the full sky. There are many varieties: with electronic sensors (CMOS or CCD) or film (especially in the past, see [1] and references therein); with monochromatic sensors or with filters, typically RGB Bayer filters but also narrower; looking to a mirror oriented to the sky or looking directly to the sky with a fish-eye lens; static cameras or moving cameras, usually installed on a sun-tracker with a shadow ball to block the direct sun light; operating at daytime, night-time or both; and others. Some of the best features of the current all-sky cameras are: they allow the possibility of changing exposure time, sensor gain and other parameters to adapt the camera to the sky scenario; they are able to obtain a snapshot of the full sky radiance, covering every sky

**Data Availability Statement:** All source code and example data files are available on Zenodo (DOI: 10.5281/zenodo.5595851).

**Funding:** This research was funded by the Ministerio de Ciencia, Innovación y Universidades

(grant no. RTI2018-097864-B-I00) and by Junta de Castilla y León (grant no. VA227P20). The funders had no role in study design, data collection and analysis, decision to publish, or preparation of the manuscript.

**Competing interests:** The authors have declared that no competing interests exist.

position and in various spectral ranges; the capture time is short; and they are inexpensive as compared to other instruments. Conversely, obtaining an accurate radiometric calibration of the images is difficult, the spectral filters are generally wide for some applications, and there are also issues with the presence of hot pixels, pixel saturation, lens aberrations, and image vignetting, among others.

This kind of instruments is generally used to observe and quantify clouds and cloud cover [2–10] or as a proxy of the sky conditions. However, all-sky cameras present high versatility and they can also be used for different purposes, among others: to derive other more complex cloud properties as cloud base height by stereoscopic methods [11, 12]; to detect and observe aurora, celestial bodies or bolides [13–15]; to estimate the sky radiance [16–18]; to study the cloud effects over solar radiation [19, 20]; to study the polarization of the sky light [21, 22]; to detect and retrieve atmospheric aerosol properties [23–25]; and to obtain synergy in combination with other instruments like radiometers, ceilometers or photometers [26–28].

The knowledge of the sky coordinates that correspond to each pixel of an all-sky camera is crucial in several applications; for example to extract sky radiance at specific sky angles [25, 28], to forecast solar irradiance [29, 30], to calculate aurora and cloud base altitudes by stereographic methods [31, 32], or simply to locate bodies over the celestial vault. This can be achieved by the geometrical calibration of the all-sky cameras, which consists of obtaining two matrices of the same size than the camera images, containing the values of both the azimuth and zenith angles viewed by each pixel. Several calibration methods have been described in the literature. The intrinsic calibration is used for determining the internal parameters of the camera. Images of a chessboard are recorded for this purpose, for instance, the OcamCalib toolbox [33, 34] employs images of the corners of the chessboard that are identified to detect any distortion in the camera optics. Extrinsic camera calibrations consist of determining the camera orientation and require the identifying the position of the Sun [30, 32, 35, 36] or any star [28, 37, 38] in several images and the correlation of these positions with the Sun or star coordinates.

We have developed the software application named ORION (all-sky camera geOmetry calibRation from star positIONs), with the main objective of providing an open and free tool for the geometrical calibration of all-sky cameras that is accurate, simple and user-friendly. The use of stars instead of the Sun for the geometric calibration is chosen because the Sun size in camera images is usually larger (causing problems to identify its center in the image) and, in addition, by using multiple stars we are able to cover more sky angles. ORION is written in Python 3 language and Qt5 for graphical interface [39], and it is capable of geometrically calibrating an all-sky camera by identifying star positions in the celestial vault. The source code and example data are hosted at Zenodo (https://doi.org/10.5281/zenodo.5595851). The Windows build of the application is hosted on the GOA-UVa (Atmospheric Optics Group, University of Valladolid) website (http://goa.uva.es/orion-app/).

This paper introduces the ORION application and it is structured as follows: Section 2 introduces the instrumentation, the workflow and the theoretical principles behind ORION while section 3 presents an example of the use of ORION. Finally, the main conclusions are summarized in Section 4.

## 2 Instrumentation, data and method

### 2.1 All-sky camera

In order to show how ORION works, sky images from an all-sky camera have been used. This camera is installed at the scientific platform of the GOA-UVa located on the rooftop of the Faculty of Sciences at Valladolid, Spain (41.664° N, 4.706° W, 705 m a.s.l.). More information

about the GOA-UVa platform and the climate conditions of Valladolid can be found in [18, 40–42].

The all-sky camera model used here is OMEA-3C from *Alcor System* manufacturer. It is formed by a SONY IMX178 RGB CMOS sensor with a fisheye lens, both encapsulated in a weatherproof case with a BK7 glass dome on top. The housing includes a heating system to avoid water condensation on the dome. This camera captures images with size of 3,096 X 2,080 pixels, a pixel scale of 5.4 arcmin/pixel and 14-bit resolution.

This all-sky camera is configured to capture a multi-exposure set of raw sky images every 2 minutes at night-time and every 5 minutes at daytime. These raw images are stored and then converted into 8-bits true color or gray-scale images. The 8-bits night-time images are used to carry out the geometrical calibration. A set of these sky images captured on August 23$^{rd}$, 2020 from 20:20 UTC to 23:58 UTC (110 images) is used for the example shown in Section 3.1.

## 2.2 Identification of stars

The data that are needed for the geometrical calibration are the position (x and y pixel coordinates) of the center of a star and the sky coordinates of the chosen star (azimuth and zenith angles). Several images obtained at different times are used in the geometrical calibration to cover a wider range of pixel positions and angles. The ORION user must put this set of images in a folder and introduce the folder path in ORION. ORION reads the date and time of each image directly from the image filename. For that, the user must prepare the image set with the following naming convention: "text_YearMonthDay_HourMinute" (example: C006_20200817_0300.jpg). ORION includes an additional utility that allows changing the file-name format of the images.

The flowchart for obtaining the data required for calibration is described in Fig 1. Two different ways of doing this have been implemented in ORION: manual and automatic mode. In automatic mode, you can select if the initial calibration matrix is Custom or Default. ORION uses these initial matrices to find the position of the pixel closest to the chosen star; for this calculation the Harvesine distance function [43] is used. Each analyzed image is first converted into gray scale.

The size of the ROI (Region of Interest) depends on the selected mode. The manual mode allows the user to choose the dimensions of the ROI. For each sky image, ORION opens an additional window where the user must select the ROI as a rectangular pixel box. To do this, ORION uses the selectROI function that is a native part of the OpenCV library [44]. To find out where to manually select the ROI, or to check whether ORION correctly identifies star positions in either Manual or Automatic mode, we recommend using Stellarium (https://stellarium.org), which is a free open source planetarium that displays a realistic hemispheric sky similar to what is captured by a fisheye lens [45]. The automatic mode skips the selection of the ROI on an image-by-image, simplifying the process. The size of the ROI box depends on the image resolution as well as the chosen option (Custom or Default).

In the "Find the star position" step, the sky coordinates for the selected star are obtained. To calculate azimuth and zenith values of a chosen star, the position of the observer (all-sky camera) is required, which is determined by the latitude, longitude, and elevation above sea level of the all-sky camera. ORION obtains the star coordinates for each image from "PyEphem" library [46], using as input the mentioned all-sky camera coordinates and the date and time when the image was captured. If the zenith angle of a star is above 83˚in an image, ORION does not consider this star for calibration in this image since it is close to the horizon, where star identification is more difficult due, in this particular case, to city lights.

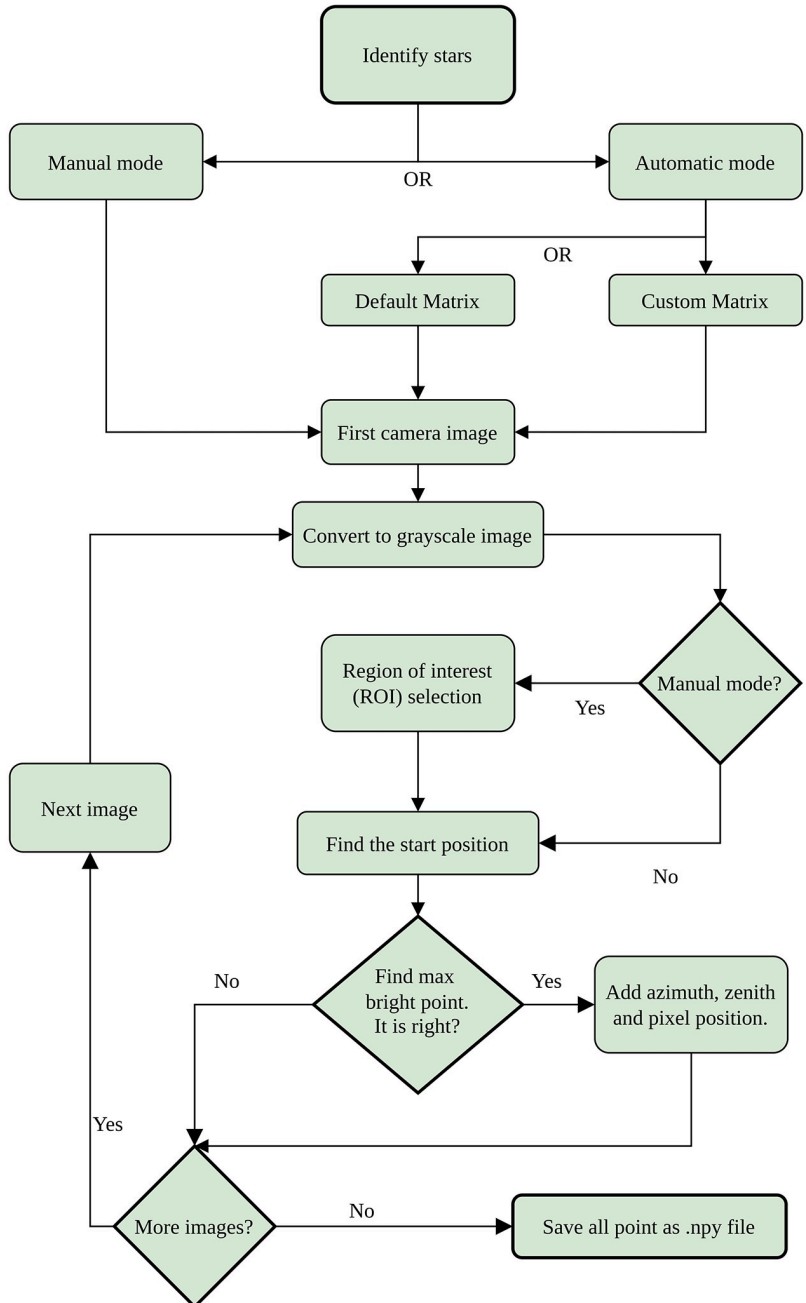

**Fig 1. Flowchart for star selection modes.** The rhombuses represent decisions made by the user.

After obtaining the coordinates of the stars, ORION only needs to identify the position of the center of the chosen star in each sky image. In manual mode ORION assumes that the center of the chosen star is located in the brightest pixel of the ROI (chosen manually) and it stores the x and y position of that pixel. In automatic mode, ORION uses the initial matrices (previously selected) to select the position of the ROI and then, to find automatically the brightest pixel inside. If the accuracy of the initial calibration matrices is admissible, the actual image of

the star in the sky image must be close to the predicted pixel (ROI is centered in this pixel), although not necessarily in the same pixel.

The Custom matrices option uses the calibration matrices that were obtained in a previous calibration, for example, using the manual mode. In this case, the ROI used to find the position of the stars is a square box whose side dimension is the height (or width if larger) of the sky image divided by 100. This arbitrary value was chosen after performing several tests and seeing empirically that it worked well for different images.

The Default option is similar to the Custom one, but the calibration matrices are generated from two input parameters instead of the matrices obtained from a previous calibration. These input parameters are: 1) the azimuth shift from North, which indicates the angle between the North of the image (assumed as the top center of the image) and the real geographical North observed in the image; and 2) the extreme zenith, which is the sky zenith angle viewed by the pixel located in the top row and center column in the image. With this information, and assuming that the center of the entire sky (zenith) corresponds to the center of the image, a calibration azimuth and zenith matrices can be calculated. Fig 2 shows a couple of examples of

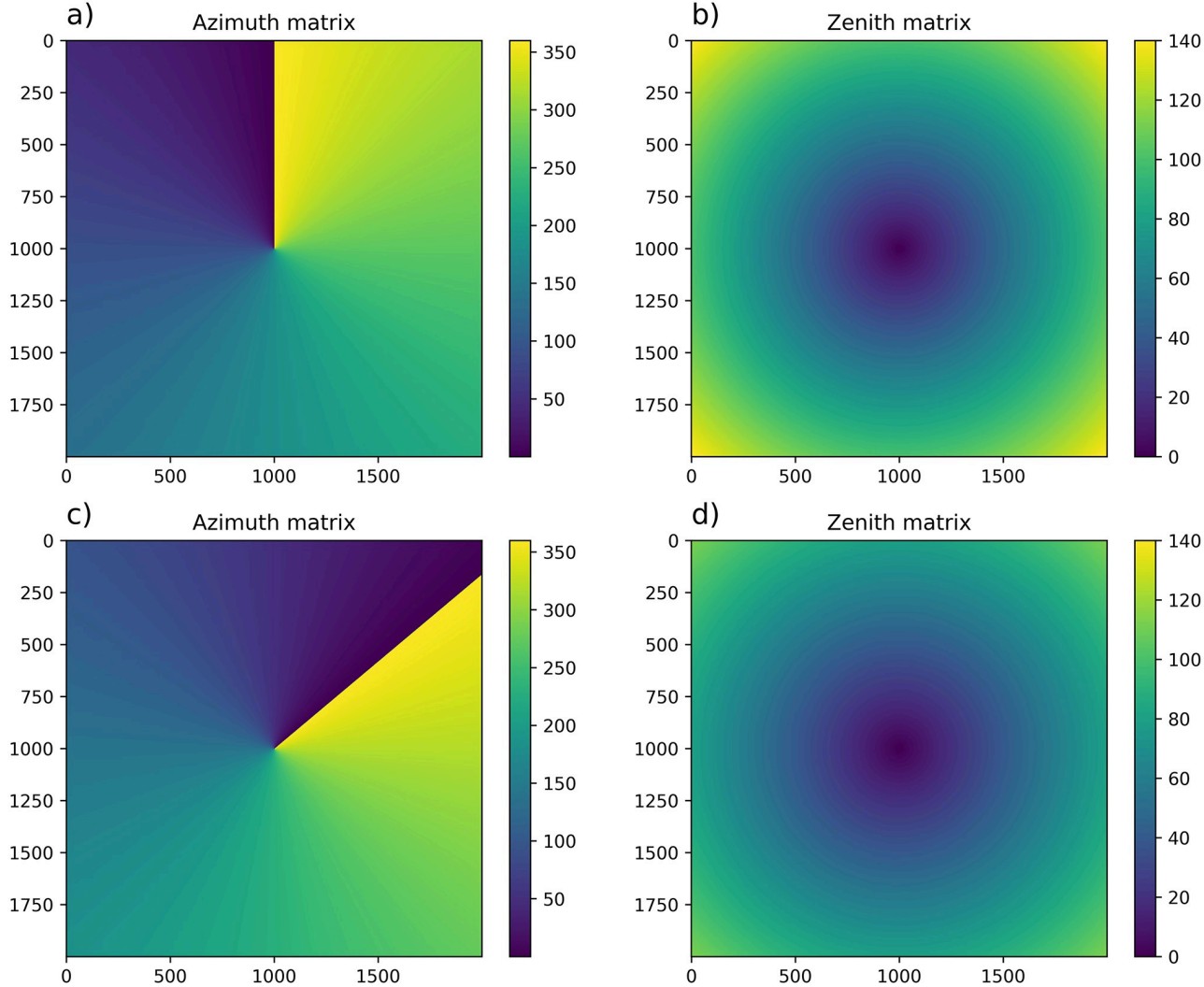

**Fig 2. Defaults matrices for azimuth and zenith.** a) Shift from north 0˚. b) Extreme zenith 100˚. c) Shift from north -50˚. d) Extreme zenith 80˚.

the matrices generated by this method in a 2,000x2,000 pixel image. The first case (Fig 2a and 2b) presents no shift from North, being the azimuth angle from 0˚to 360˚counterclockwise, and a high range of zenith angles; while the second case (Fig 2c and 2d) presents a well defined shift from North of -50˚and a lower variation in zenith angles due to the lower value of the extreme zenith angle. Finally, the ROI used to find the stars in Default option is a square box similar to that in Custom mode, but dividing by 50 instead of 100 (i.e. larger box), since the Default method is less accurate. This mode is useful to perform a quick calibration as an input to the automatic star detection mode.

The user must decide if the position of the brightest pixel is correct. If yes, the azimuth, zenith and the pixel position (x, y) will be stored and ORION will go to the next image. This process is repeated until the list of selected images is complete. This data is stored in an *.npy* file for use in calibration. More files can be generated for other stars.

## 2.3 Calibration algorithm

The position of a pixel in an image is given in Cartesian coordinates, where x and y coordinates are the column and row numbers, respectively, and the system is centered in the left-top pixel (x = 0, y = 0). The first element corresponds to zero position for x and y because ORION is programmed in Python 3 language. It is convenient to convert this system into polar coordinates, which more directly corresponds to the zenith and azimuth angles in the sky. In this sense, a polar coordinate system centered in the zenith of the sky with Cartesian coordinates equal to x = $x_C$ and y = $y_C$, can be described as:

$$r = \sqrt{\left(x_C - x\right)^2 + \left(y_C - y\right)^2} \tag{1}$$

$$\Phi = \arctan\left(\frac{x_C - x}{y_C - y}\right) \tag{2}$$

where r is the radial distance (in pixels) from the center, and $\Phi$ is the polar angle in this new system. This polar angle presents its zero value in the direction of the y-axis, in a similar way as in Fig 2a.

Radial distance and polar angle are directly related to the sky zenith and azimuth angles, respectively. Assuming polar symmetry and that the camera is well leveled, the polar angle must be equal to the sky azimuth with a shift due to a non perfect alignment of the camera with respect to the North. This shift is the same used in the Default option of Automatic mode in Section 3.1. Sky zenith angle is related to radial distance, and the relationship between both can be linear (which is assumed in the mentioned Default option), or a higher degree polynomial. Hence, if we determined the $x_C$ and $y_C$ coordinates, we will obtain the polar coordinates from Eqs 1 and 2. Then, if we determine the shift of the polar angle with respect to the sky azimuth, and the relationship between radial distance and zenith angle, we can transform the obtained polar coordinates into the sky angles viewed by each pixel. This is the way that ORION uses to calculate the calibration matrices.

The required $x_C$ and $y_C$ values, the azimuth shift from polar coordinate and the relationship between zenith angle and radial distance can be obtained from the stored data in the previous Section 2.2: real azimuth and zenith star values and x and y pixel positions where the stars were found.

First, ORION calculates the $x_C$ and $y_C$ values. It involves four iterations to obtain a better precision after each iteration. Each iteration consists of scanning different pixel coordinates and assuming that they are the $x_C$ and $y_C$ values. The first iteration assumes that $x_C$ and $y_C$

values must be near to the center of the image itself $(x_{ci}, y_{ci})$: $x_{ci}$ = (width-1)/2 and $y_{ci}$ = (height-1)/2. A scan from x = $x_{ci}$ -250 to x = $x_{ci}$+250 in 5 pixel steps is done. For each scan in x-column, an additional scan in y-raw is done from x = $y_{ci}$ -250 to x = $y_{ci}$+250 in 5 pixel steps too. It implies that ORION tests 101x101 positions (10,201) in this iteration for the $x_C$ and $y_C$ values. For each one of these 10,201 potential centers, ORION calculates the Φ values by Eq 2 for all star positions that were chosen and previously stored, and then calculates the difference between these Φ values and the real azimuth of the stars (this information was also stored). If the center of the image is well determined, these differences must be constant and equal to the mentioned shift angle. Then, ORION calculates the standard deviation of these differences; the x and y position among the 10,201 showing the lowest standard deviation is assumed as the true $x_C$ and $y_C$ values.

This first iteration provides a good approximation of the real $x_C$ and $y_C$ values; however, the precision can still be improved, since the scans were done every 5 pixels in order to encompass a large part of the image but expending low computation time. Three more iterations are done by assuming the result of the previous iteration as the initial conditions. The second iteration scans from -50 to 50 around $x_{ci}$ and $y_{ci}$ in 1 pixel step; while the third and fourth scan from -0.5 to 0.5 and from -0.005 to 0.005 in 0.01 and 0.001 pixel steps, respectively. After the four iterations, ORION stops and provides the $x_C$ and $y_C$ values with a precision of 0.001 pixels.

Once $x_C$ and $y_C$ are calculated, the azimuth shift is calculated by ORION using the stored star information. A linear fit between the polar angle obtained using the calculated $x_C$ and $y_C$ and the real azimuth of the stars in the sky is performed by ORION. The y-intercept is considered the azimuth shift angle. This linear fit is done excluding star positions with zenith angles below 20˚, in order to avoid pixels for which a little variation in x and y position implies a big variation in the polar angle.

Once the azimuth shift and the $x_C$ and $y_C$ values are calculated, ORION calculates the radial distance using the stored star information and Eq 1. A polynomial fit between the stored real zenith angle of the stars and the obtained radial distance of the image is performed. The degree of this polynomial fit is chosen by the user. After that, the zenith angle viewed by each pixel can be directly obtained by applying the fit coefficients to the known radial distance, which is also available from the previously calculated $x_C$ and $y_C$ values. This information is enough to obtain the geometrical calibration matrices of sky azimuth and zenith angles.

## 3 Results

### 3.1 Star position detection and calculation of azimuth and zenith matrices

An example of ORION calibration with real images is shown in this section. The used images correspond to the all-sky camera described in Section 2.1, installed at Valladolid. We selected the images every 2 minutes from 20:20 UTC to 23:58 UTC on August 23[rd], 2020. We chose this set of images since it corresponds to a clear night with fully cloudless conditions and no Moon. It is however possible to use moonlight nights and even partly cloudy nights, as long as there are visible stars. First of all, the camera location information and the local path for image folder are introduced in the input parameters. After that, we can choose between manual or automatic mode (see Fig 1). Automatic default option is selected since we have no previous custom calibration. The information about default matrices in this example is: the shift from north equal to -5˚and extreme zenith equal to 95˚. Both parameters are introduced in the application as can be observed in the screenshot of Fig 3.

Once this information has been entered, we start to identify and detect the position of the stars in the sky images. The first star chosen in this example is Capella. This star is selected from the list of available stars. The path where the data file will be stored is an input. The

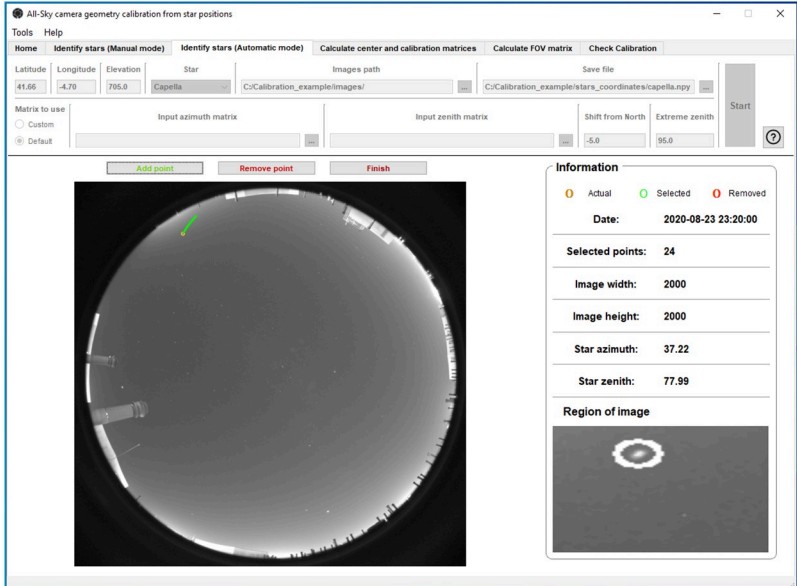

**Fig 3. ORION screenshot of the star identification process in automatic mode under default matrices option.** The chosen star is Capella.

procedure described here is the same for each star. ORION obtains, from the "PyEphem" library, the azimuth and zenith angles corresponding to the time and place of the first image; then the position of the pixel closest to the coordinates of the star is obtained following the calibration matrices given by the Default input parameters. The position of the brightest pixel inside the ROI (a square box centered on the mentioned "nearest pixel") is considered as the position of the center of the star chosen in the image. ORION displays the position of the chosen star marked with a white circle in the main sky image (left part of the application; Fig 3) as well as in an enlarged image in the bottom-right corner of the application ("Region of Image" in Fig 3). This enlarged image is useful to discern whether the position of the chosen star is correct or not (hot pixels can lead to erroneous identification). Considering that the position of the star is well identified, the data is stored and we can continue with the next image. Correctly selected points are displayed in the image in green, wrong points are marked in red.

When a star position is added or removed (either in Manual or Automatic mode), then ORION analyzes the next sky image (see Fig 1) and repeats the process until all the available images are analyzed. Once it is finished, ORION stores in the chosen file (in this case "capella.npy") the azimuth and zenith angles of the star and the x and y pixel positions assigned to this star in each sky image. This file contains the necessary information for the calculation of the calibration matrix. However, a single star in one night does not usually cover the full range of zenith and azimuth angles. Therefore, it is recommended to perform the calibration with the positions of several stars.

In this example, we repeat the process to obtain the positions of Capella, Altair, Vega and Deneb. Once the four files are generated with the Default mode, they are used to obtain the azimuth and zenith calibration matrices of the camera. ORION calculates the image center and the relationship between the zenith angle and the radial distance, as explained in Section 2.3, from the previous files (star positions and coordinates) chosen by the user. The degree of the polynomial fit between zenith angle and radial distance can be chosen, being equal to 2 in our example (Fig 4), since the 1st degree option (linear fit) does not fit to the data under high

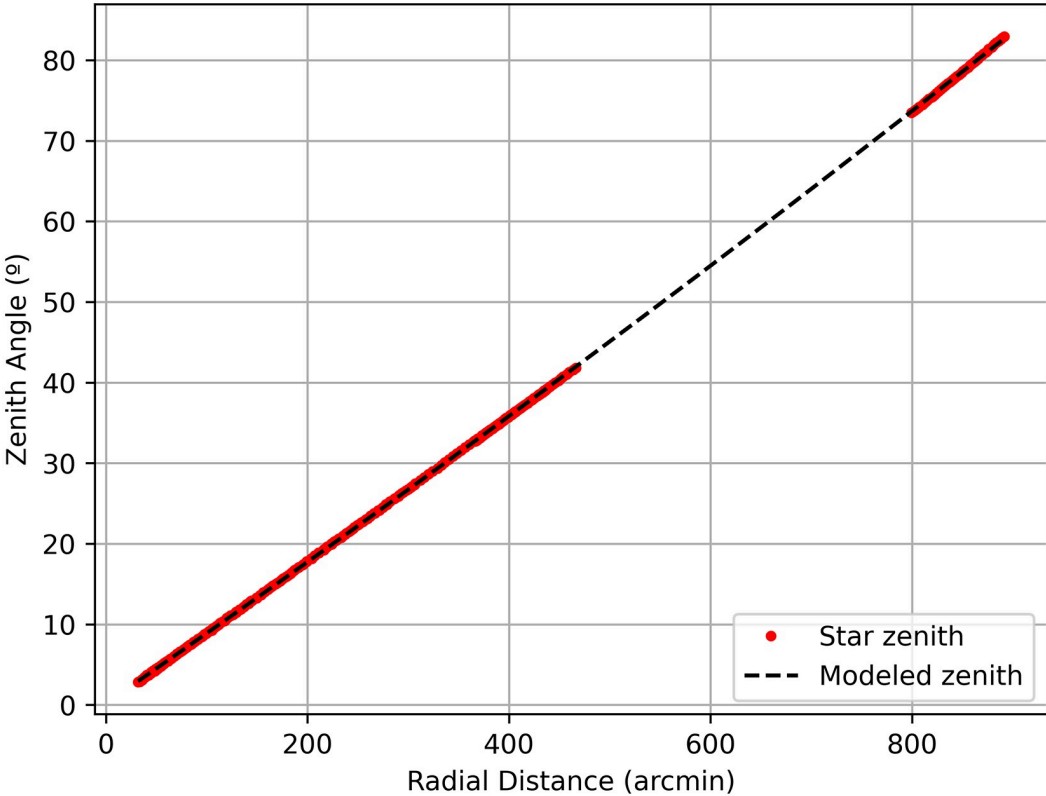

**Fig 4. Relationship between zenith angle and radial distance with polynomial adjustment of degree 2.**

zenith angle values (see S1 Fig). In this case the center of the image is (1000.148, 1000.341) and an offset with respect to north of -5.3˚. It can be seen in Fig 4 that there were no data for zenith angles between 40 and 70 degrees. It is recommended to cover the maximum range of zenith angles for calibration, therefore in this example we need to look for more star positions to fill the zenith angle gap. For this purpose, two additional stars are selected: Arcturus and Alphecca.

The pixel positions of Arcturus and Alphecca are retrieved using Automatic mode but Custom option. This option must be more accurate than the Default. This mode is equal to Default but using a pre-calculated azimuth and zenith matrices (the path is introduced as input, see Fig 5) and considering a smaller ROI (square box with side 100 times smaller than the maximum dimension of the analyzed sky image, see Section 2.2) for finding the brightest pixel. Fig 5 presents the calculation of pixel and star positions for Alphecca. In this case, some pixels have not been correctly identified by ORION (the star was mixed with a hot pixel), and those are marked with red circles. The positions of these pixels and the coordinates of the star are not stored for these cases.

In this example, the calibration matrices are recalculated adding the information of these two new stars. In Fig 6a we have the azimuth calibration matrix and in Fig 6b, the zenith matrix. As a final result, the center of the sky image, which corresponds to the sky zenith, is really close to the center of the sky image. Regarding the shift from North, it is -5.23˚. Fig 6c shows that the fit between zenith angles and radial distance is now performed with a full range of zenith angles, and the second-order polynomial fit can be better observed. In addition, zenith angles from 2˚ to 82˚ have been covered.

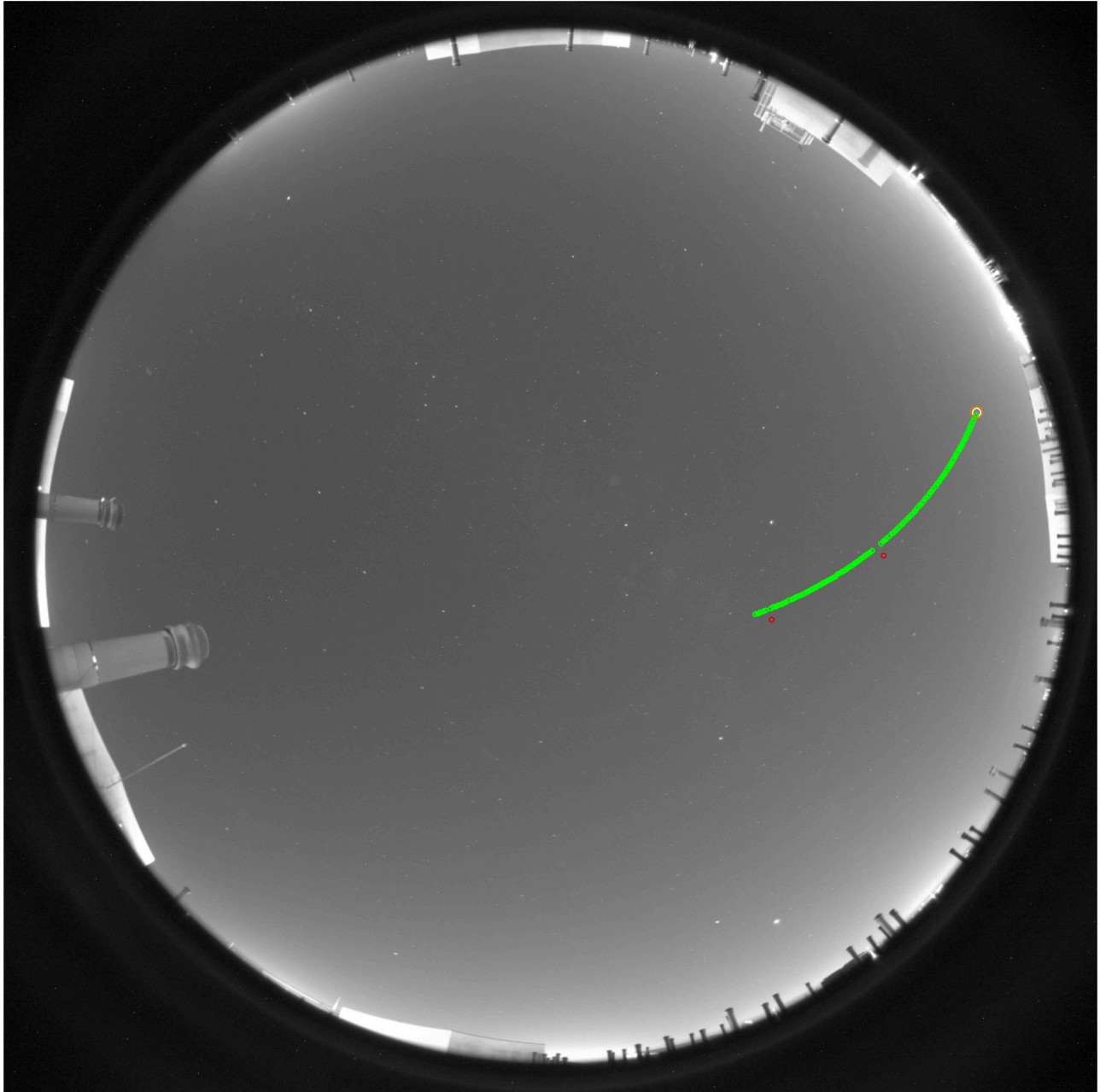

**Fig 5. Selected (green dots) and discarded (red dots) points in the image sequence for the Alphecca star.**

## 3.2 Calibration check

Once the final calibration matrices have been calculated, their performance for the detection of stars in a sky image can be evaluated with the "Check calibration" functionality. For this purpose, one star must be chosen, for example Alioth, which has not be used in the calibration process, as shown in Fig 7. ORION analyzes each single image from the set in the image path. The calibration matrices point out that the center of the chosen star should be in a certain pixel (marked in red circle in Fig 7). ORION also looks for the brightest pixel in a box centered

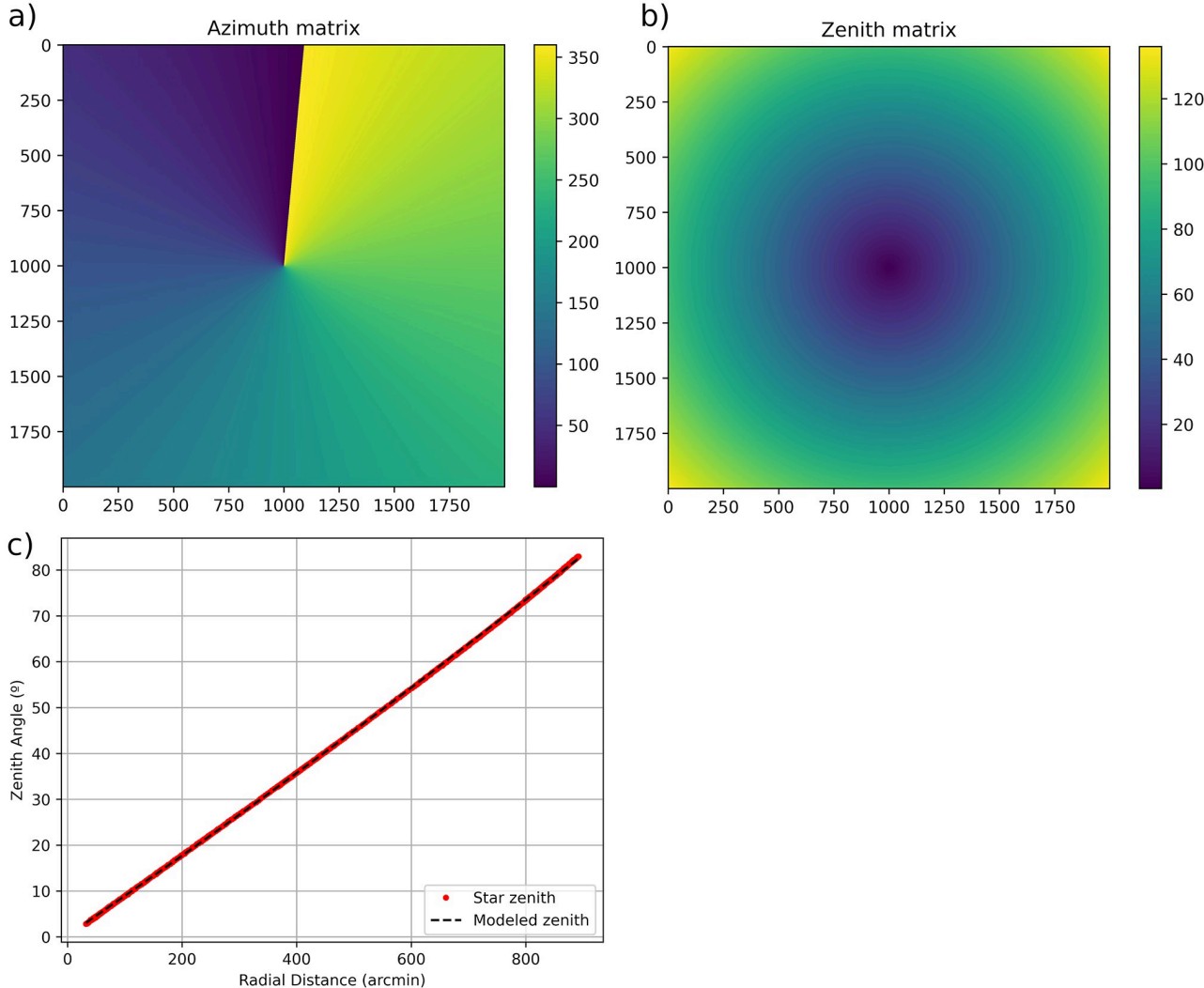

**Fig 6. Calibration result: a) azimuth matrix, b) zenith matrix, c) ratio of zenith angle to radial distance with polynomial adjustment of degree 2.**

in the mentioned pixel. This box has the size of the box used in the Default option of Automatic mode (see section 2.2). The brightest pixel (marked as green circle; see Fig 7) is assumed as the real position of the star center in the analyzed sky image.

The software calculates the real star position (brightest pixel) and the one predicted by calibration matrices for all images, and then uses both positions to quantify the agreement between the predicted star position by calibration and the real one. Five panels with different analyses are provided for this verification, see Fig 8.

Fig 8a, which is the default panel shown by ORION (see Fig 7), represents the pixel distance between the predicted and real star positions for each image in the analyzed set. This distance is between 0 and 15 arcmin in the analyzed example, being the mean distance about 7.25 arcmin (dotted red line). This means that the obtained calibration matrices predicted the center of the star Alioth with an average difference of 7.25 arcmin (about 1.5 pixels) with respect to the real position. These differences are also presented as a function of the star azimuth (Fig 8b) and zenith (Fig 8c) angles. No azimuth or zenith angle dependence is observed in the analyzed

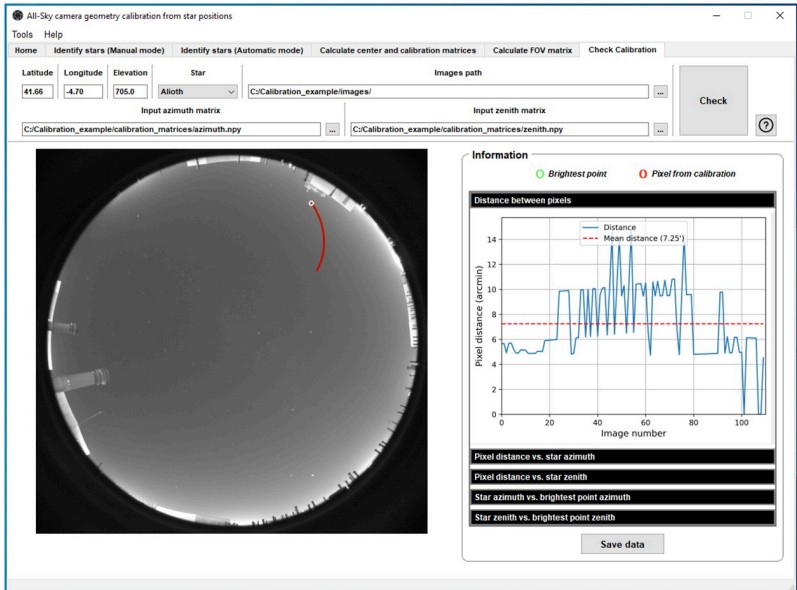

**Fig 7. ORION screenshot for checking the calibration feature using the star Alioth.**

example. Finally, ORION shows the azimuth (Fig 8d) and zenith (Fig 8e) angles, given by the calibration matrices, of the real star positions in the image (brightest pixel) as a function of the real ones angles for all analyzed sky images. As it can be observed in the example, the angles assigned by the calibration matrices to the star position in the image (brightest pixel) highly correlate with the real coordinates of the star.

This calibration check has been also carried out for 9 additional stars, see Table 1. It is observed that the mean for all stars ranges from 4 to 12 arcmin. The best results are obtained by Kochab with a mean of 4.50 arcmin and a standard deviation of 3.24 arcmin. The highest values are observed in Dubhe, Mirach and Sirrah, which correspond to the presence of hot pixels in positions close to the stars, that are not well identified. In general, the mean accuracy for the 10 stars is about 9.0 arcmin (1.7 pixels) and the mean precision, given by the standard deviation, is about 7.5 arcmin (1.4 pixels).

## 4 Conclusions

This paper presents ORION, a new software application, which provides the geometrical calibration of all-sky cameras using a set of sky images captured at night-time under cloudless conditions. An example of use has shown the capability of this application to obtain the azimuth and zenith angles viewed by each pixel of the camera. The accuracy of the calibration depends on the chosen stars and the sky positions covered by them. This accuracy can be also checked with ORION itself. A simple calibration was able to estimate the star positions with an average accuracy about 9.0 arcmin in the provided example, using a camera with 5.4 arcmin/ pixel resolution (1.7 pixels). The average precision (standard deviation) in this case is about 7.5 arcmin (1.4 pixels). We encourage other users and researchers to use the ORION application for easy geometrical calibration of all-sky cameras, which will be helpful to locate any body (stars, planets, Sun, Moon, among others) in their sky images, if the sky coordinates of that body are known. Moreover, ORION includes other features such as exporting data in different

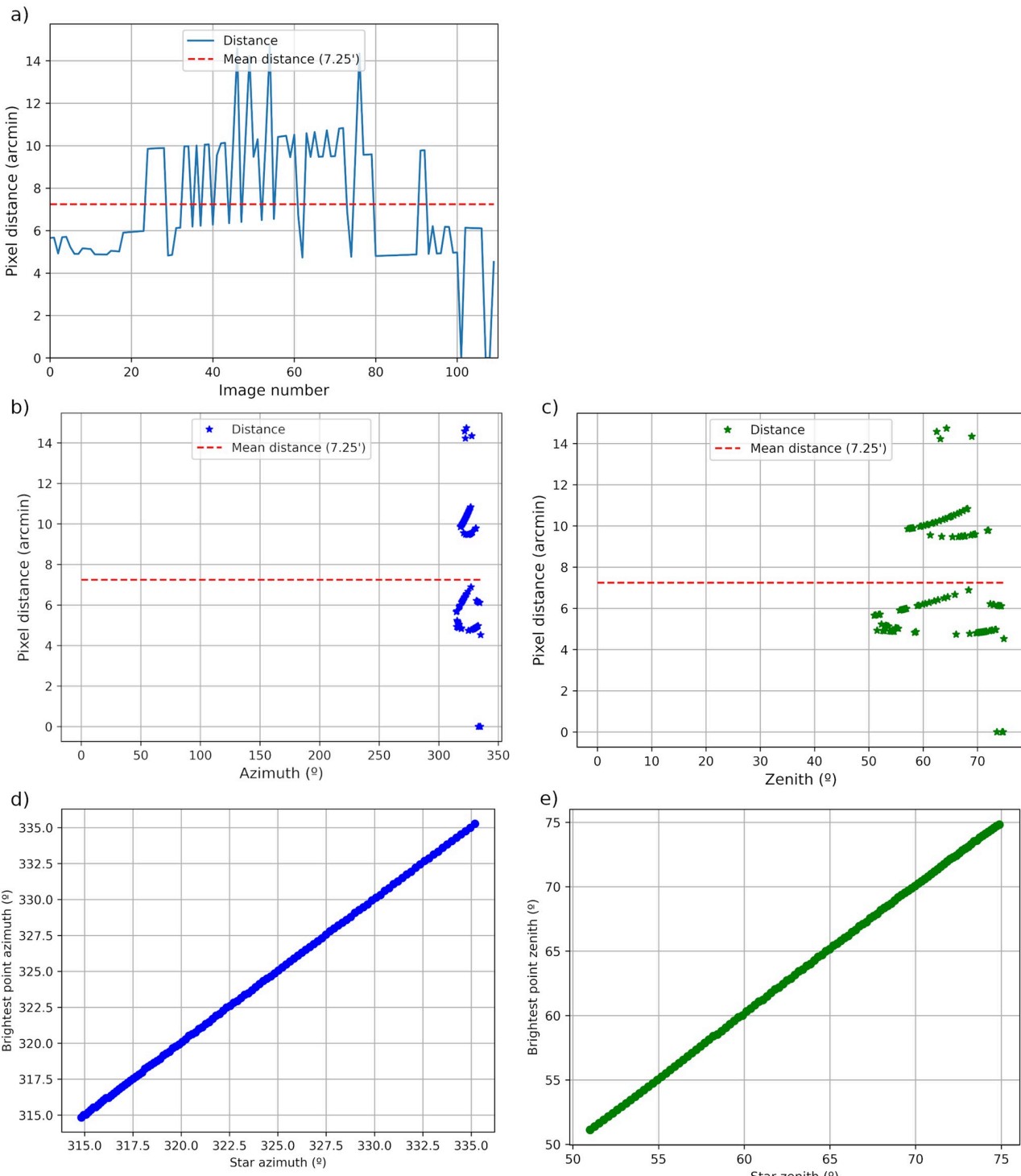

**Fig 8.** Performance of the obtained calibrations matrices for Alioth star positions: a) azimuth obtained with the calibration for the located star position (brightest point) as a function of the real star azimuth; b) zenith obtained with the calibration for the located star position (brightest point) as a function of the real star zenith; c) absolute pixel distance between the star position given by the calibration matrices and the position of the assumed real center of the star in the image (brightest point in a square box defined by the height or width of the sky image) as a function of star azimuth angle (red dotted line represents the mean absolute distance); d) absolute pixel distance between the star position given by the calibration matrices and the position of the assumed real center of the star in the image (brightest point in a square box defined by the height or width of the sky image) as a function of star zenith angle (red dotted line represents the mean absolute distance); e) absolute pixel distance between the star position given by the calibration matrices and the position of the assumed real center of the star in the image (brightest point in a square box defined by the height or width of the sky image) for each available image (red dotted line represents the mean absolute distance).

**Table 1. Calibration check for others stars.**

| Star | N | Min (arcmin) | Max (arcmin) | Mean (arcmin) | Std (arcmin) |
|---|---|---|---|---|---|
| *Algol* | 89 | 4.27 | 29.66 | 11.97 | 5.40 |
| *Alioth* | 110 | 0 | 14.74 | 7.25 | 2.87 |
| *Antares* | 53 | 0 | 21.17 | 9.05 | 4.76 |
| *Dubhe* | 110 | 0 | 119.0 | 10.06 | 16.74 |
| *Fomalhaut* | 59 | 0 | 30.66 | 12.0 | 8.05 |
| *Kochab* | 110 | 0 | 11.33 | 4.50 | 3.24 |
| *Mirach* | 110 | 0 | 74.07 | 10.30 | 10.70 |
| *Mizar* | 110 | 0 | 14.20 | 6.63 | 2.87 |
| *Shedar* | 110 | 0 | 12.37 | 7.54 | 2.75 |
| *Sirrah* | 110 | 0 | 121.58 | 11.12 | 17.42 |

Number of points (N), minimum (Min), maximum (max), mean (Mean) and standard deviation (Std) of the calibration check for different stars.

formats, or calculating the field of view of each pixel, which is not detailed in this paper but can be useful for the ORION users.

## Supporting information

**S1 Fig. Relationship between zenith angle and radial distance with polynomial adjustment of degree 1.**
(TIFF)

## Acknowledgments

The authors gratefully thank AERONET for the aerosol products used. Finally, the authors thank the GOA-UVa staff members Rogelio Carracedo, Daniel González-Fernández, Sara Herrero and Patricia Martín, who helped with the operation and maintenance of the camera.

## Author Contributions

**Conceptualization:** Juan Carlos Antuña-Sánchez, Roberto Román.

**Data curation:** Juan Carlos Antuña-Sánchez, Ramiro González.

**Formal analysis:** Roberto Román.

**Funding acquisition:** Victoria Cachorro, Ángel de Frutos.

**Methodology:** Juan Carlos Antuña-Sánchez, Roberto Román, Juan Luis Bosch.

**Project administration:** Victoria Cachorro, Ángel de Frutos.

**Resources:** Carlos Toledano, David Mateos, Ramiro González.

**Software:** Juan Carlos Antuña-Sánchez, Roberto Román.

**Supervision:** Roberto Román, Carlos Toledano.

**Validation:** Juan Carlos Antuña-Sánchez, Roberto Román.

**Visualization:** Juan Carlos Antuña-Sánchez.

**Writing – original draft:** Juan Carlos Antuña-Sánchez, Roberto Román.

**Writing – review & editing:** Juan Luis Bosch, Carlos Toledano, David Mateos, Ramiro González, Victoria Cachorro, Ángel de Frutos.

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
