## [Decision Letter · Decision Letter 0]

29 Nov 2021

PONE-D-21-34079ORION software tool for the geometrical calibration of all-sky camerasPLOS ONE

Dear Dr. Antuña-Sánchez,

Thank you for submitting your manuscript to PLOS ONE. After careful consideration, we feel that it has merit but does not fully meet PLOS ONE’s publication criteria as it currently stands. Therefore, we invite you to submit a revised version of the manuscript that addresses the points raised during the review process.

The material in the paper looks interesting and sound, although the manuscript should be revised carefully to meet PLOS ONE publication criteria. 

The authors should put an effort to improve the paper by taking into account the comments of all the referees, in particular those by Reviewer 1 who raised important remarks particularly related to the methodology used for the computational experiments, and to the presentation and the use of English language in the paper. Some sentences are not easy to read, and some typos are still present at places. It would help if you had a proof check of the English by a professional. Overall the authors should carefully check the English of the paper more deeply. 

Furthermore, the scientific principles described in the paper look sound, but more statistical analysis should be performed as suggested by the reviewers to enhance the manuscript.

Please take carefully into account the comments of all the referees for improving the manuscript to meet PLOS ONE standards before resubmitting it to the journal.

We look forward to receiving your revised manuscript.

Kind regards,

Sergio Consoli

Academic Editor

PLOS ONE

Journal Requirements:

"This research was funded by the Ministerio de Ciencia, Innovaci´on y Universidades (grant no. RTI2018-097864-B-I00) and by Junta de Castilla y Le´on (grant no. VA227P20). "

"No, The funders had no role in study design, data collection and analysis, decision to publish, or preparation of the manuscript."

Reviewers' comments:

Reviewer's Responses to Questions

**Comments to the Author**

1. Is the manuscript technically sound, and do the data support the conclusions?

Reviewer #1: Yes

Reviewer #2: Yes

2. Has the statistical analysis been performed appropriately and rigorously? 

Reviewer #1: Yes

Reviewer #2: Yes

3. Have the authors made all data underlying the findings in their manuscript fully available?

Reviewer #1: Yes

Reviewer #2: Yes

4. Is the manuscript presented in an intelligible fashion and written in standard English?

Reviewer #1: No

Reviewer #2: Yes

5. Review Comments to the Author

Reviewer #1: COMMENTS

This paper is on developing an open-source software for geometric calibration of all-sky cameras. This is an interesting application for publication in PLOS ONE. Authors argue that knowledge about the sky coordinates represented in each pixel of an all-sky camera is important for many applications.

In the introduction section, the authors give useful background about the all-sky cameras and geometric calibration which is the topic of the current paper. They introduce ORION to fill the gaps about geometric calibration of the sky images. A relevant question is, is there any other software (open access/paid) which can be used for calibration, over which this software is a major advancement?

The authors describe the instrumentation, the workflow and the theoretical principles behind ORION adequately in Section 2. In Section 3, the authors present the use of ORION.

Specific comments:

Numerous places where writing style cannot be easily understood has been corrected for in the manuscript. (See the notes in the PDF). Additional comments/suggestions are suggested below.

Rewrite L33-36. Rewrite as per the suggestion in the PDF.

Rewrite L37-40 something like “Extrinsic camera calibrations consists of determining camera orientation and requires dentification of the positions of the Sun [32, 35, 36] or any star [28, 37, 38] in several images and the correlation of these positions with the Sun or star coordinates.”

Rewrite L64 as “It is formed by a SONY IMX178 RGB CMOS sensor with a fisheye lens, both encapsulated in a weatherproof case with a BK7 glass dome on top.”

Rewrite L66-67 as “The camera can capture image with size of 3096 X 2080 pixels, a pixel scale of 5.4 arcmin/pixel and 14-bit resolution”.

L79-81- It will be useful to show the mathematical equation to calculate azimuth and zenith value of star from the information about the observer.

Rewrite L87-90 “As several images obtained at different times is used in the geometrical calibration to cover a wider range of pixel positions and angles, the ORION user must put this image set in a folder and introduce the folder path in ORION.”

L124-L125. Can authors present some statistical analysis (maybe in a Supplementary section) of how the accuracy varies with the size of the square box. The idea is to provide some convincing evidence of assuming this value in the software. I assume the users cannot change this value when theyb are turning the software.

More description about the image 1 is needed. Example, what does “Find the start position mean”. What does “Find Max Bright point. Is it right?” mean. I believe that an image is discarded if we cannot find the bright point.

L235-236 – The authors mention that they chose the image not having moon. I believe their analysis and software would not be useful if the images have moon?4

Overall impression

The scientific principles described in this paper are sound, however more statistical analysis has been suggested in a few places which can enhance the manuscript. Further, the language and writing style makes the MS difficult to read and make sense of the meaning in many places. Some of these mistakes have been corrected in the main text, however it is recommended that the author gets the manuscript checked and proofread by a native English speaker or by a professional editing service who can put the sentences in order to enhance the understanding. A major revision is suggested for the authors before the manuscript can be accepted. The authors should be more aware of scientific writing versus the free style writing which has been used in the MS.

Reviewer #2: This paper developed a new open-source software, ORION, to conduct geometrical calibration to all-sky cameras under restricted conditions (e.g., cloudless). From the writing perspective, it is good to provide the principles underlying the software and an example to show how it works. However, I think there are too many operation-level description (e.g., what button to click in this software), which makes some part of the paper more like a user guide. It would be improved if the authors deliver it from a perspective of an academic study instead of from a user guide scope (see detail in section-wise comments). A few issues about formats (e.g., thousand separator) and grammar need to be double checked (see detail in line-wise comments).

Section-wise comments:

2.4 Too detail in terms of the techniques (e.g., different forms of output). It makes it more like a user guide not academic study

3 Some operations may not be needed because that is more like a user-guide. For example, you mentioned what will happen after you click a specific button (e.g., “Start” and “Check” buttons).

Figure 1 You may want to include a legend for different shapes in your flow chart.

Figures 3-9 I do not suggest using screenshots of the software to deliver the functions because that makes it like a user guide. You may simply separate different functions or different stages of processing to different sections in your paper and simply visualize the results of each stage (or functions) you get, like Figure 2.

Line-wise comments

47 python3->Python 3

47 please give a reference link to Qt5. I am not sure what it is.

59 This is the second time you mention GOA-UVa but your first time mentions the full name. You may want to mention it in line 51

67 3096 X 2080 -> 3,096 X 2,080. You may need thousand separators.

77 a space before “To”?

157 You may want no indent before “where”

183 10,201thousand separators

184 10,201thousand separators

190 10,201thousand separators

338 difference -> different?

6. PLOS authors have the option to publish the peer review history of their article (what does this mean?). If published, this will include your full peer review and any attached files.

Reviewer #1: No

Reviewer #2: **Yes: **Tianyang Chen

---

## [Author Response · Author response to Decision Letter 0]

13 Feb 2022

Dear Editor,

We appreciate the work done in revising our manuscript. English has been revised throughout the paper, taking into account the reviewers' comments and yours. In addition, section 2.2 has been rewritten and the description of Figure 1 has been improved. The figures have been reworked, removing some screenshots. The responses to the reviewers are presented below:

Response to the Reviewer #1 

First, we are grateful for the effort of Referee #1 and her/his review in detail. Reviewer comments (RC), and author comments (AC)

Reviewer #1:

RC: This paper is on developing an open-source software for geometric calibration of all-sky cameras. This is an interesting application for publication in PLOS ONE. Authors argue that knowledge about the sky coordinates represented in each pixel of an all-sky camera is important for many applications. In the introduction section, the authors give useful background about the all-sky cameras and geometric calibration which is the topic of the current paper. They introduce ORION to fill the gaps about geometric calibration of the sky images. A relevant question is, is there any other software (open access/paid) which can be used for calibration, over which this software is a major advancement?

AC: We have mentioned other toolboxes and methodology for geometric calibration of all-sky cameras (line 40-43). In the case of ORION we have developed a multiplatform application, with a graphical interface that makes it easier for the user to select the data and calibration. In addition, field of view calculation and calibration validation have been integrated. 

The authors describe the instrumentation, the workflow and the theoretical principles behind ORION adequately in Section 2. In Section 3, the authors present the use of ORION.

Specific comments:

RC: Numerous places where writing style cannot be easily understood has been corrected for in the manuscript. (See the notes in the PDF). Additional comments/suggestions are suggested below.

AC: These corrections have been applied.

RC: Rewrite L33-36. Rewrite as per the suggestion in the PDF.

AC: Done.

RC: Rewrite L37-40 something like “Extrinsic camera calibrations consists of determining camera orientation and requires dentification of the positions of the Sun [32, 35, 36] or any star [28, 37, 38] in several images and the correlation of these positions with the Sun or star coordinates.”

AC: It has been rewritten.

RC: Rewrite L64 as “It is formed by a SONY IMX178 RGB CMOS sensor with a fisheye lens, both encapsulated in a weatherproof case with a BK7 glass dome on top.”

AC: Done.

RC: Rewrite L66-67 as “The camera can capture image with size of 3096 X 2080 pixels, a pixel scale of 5.4 arcmin/pixel and 14-bit resolution”.

AC: It has been rewritten.

RC: L79-81- It will be useful to show the mathematical equation to calculate azimuth and zenith value of star from the information about the observer.

AC: The PyEphem library is used to calculate the azimuth and zenith, which "generates positions using techniques from the 1980s popularized in Jean Meeus' Astronomical Algorithms, such as the IAU's 1980 Earth nutation model and the VSOP87 planetary theory." (https://rhodesmill.org/pyephem/). The details of such algorithms are out of the scope of our manuscript.

RC: Rewrite L87-90 “As several images obtained at different times is used in the geometrical calibration to cover a wider range of pixel positions and angles, the ORION user must put this image set in a folder and introduce the folder path in ORION.”

AC: It has been Rewritten.

RC: L124-L125. Can authors present some statistical analysis (maybe in a Supplementary section) of how the accuracy varies with the size of the square box. The idea is to provide some convincing evidence of assuming this value in the software. I assume the users cannot change this value when they are turning the software.

AC: This reviewer comment is really interesting. This value was chosen empirically after several tests that we performed and it was found to work best for different images. In the next version of ORION the option to change the box size will be added.

RC: More description about the image 1 is needed. Example, what does “Find the start position mean”.What does “Find Max Bright point. Is it right?” mean. I believe that an image is discarded if we cannot find the bright point.

AC: Section 2.2 has been rewritten to describe step by step the flowchart (Figure 1), taking into account the proposed corrections.

RC: L235-236 – The authors mention that they chose the image not having moon. I believe their analysis and software would not be useful if the images have moon?

AC: We added this sentence: “It is however possible to use moonlight nights and even partly cloudy nights, as long as there are visible stars.”

Overall impression

RC: The scientific principles described in this paper are sound, however more statistical analysis has been suggested in a few places which can enhance the manuscript. Further, the language and writing style makes the MS difficult to read and make sense of the meaning in many places. Some of these mistakes have been corrected in the main text, however it is recommended that the author gets the manuscript checked and proofread by a native English speaker or by a professional editing service who can put the sentences in order to enhance the understanding. A major revision is suggested for theauthors before the manuscript can be accepted. The authors should be more aware of scientific writing versus the free style writing which has been used in the MS. 

AC: We have reviewed the language and writing style in this new version.

Response to the Reviewer #2

First, we are grateful for the effort of Referee #2 and her/his review in detail. Reviewer comments (RC), and author comments (AC).

Reviewer #2:

RC: This paper developed a new open-source software, ORION, to conduct geometrical calibration to all-sky cameras under restricted conditions (e.g., cloudless). From the writing perspective, it is good to provide the principles underlying the software and an example to show how it works. However, I think there are too many operation-level description (e.g., what button to click in this software), which makes some part of the paper more like a user guide. It would be improved if the authors deliver it from a perspective of an academic study instead of from a user guide scope (see detail in section-wise comments). A few issues about formats (e.g., thousand separator) and grammar need to be double checked (see detail in line-wise comments).

Section-wise comments:

RC: Section 2.4 Too detail in terms of the techniques (e.g., different forms of output). It makes it more like a user guide not academic study.

AC: This section has been removed, as it is descriptive of additional utilities of the application.

RC: Section 3 Some operations may not be needed because that is more like a user-guide. For example, you mentioned what will happen after you click a specific button (e.g., “Start” and “Check” buttons).

AC: This kind of statements has been corrected in the new manuscript.

RC: Figure 1 You may want to include a legend for different shapes in your flow chart.

AC: The arrowheads in the flowchart have been fixed. In the description of Figure 1 the meaning of the rhombuses has been added with the following sentence "The rhombuses represent decisions made by the user."

RC: Figures 3-9 I do not suggest using screenshots of the software to deliver the functions because that makes it like a user guide. You may simply separate different functions or different stages of processing to different sections in your paper and simply visualize the results of each stage (or functions) you get, like Figure 2.

AC: Figure 4 has been removed and Figures 5, 6, 7 have been substituted to improve the visualization of the results following the reviewer comment. 

Line-wise comments

RC: L47 python3->Python 3

AC: It has been changed.

RC: L47 please give a reference link to Qt5. I am not sure what it is.

AC: We have added a reference.

RC: L59 This is the second time you mention GOA-UVa but your first time mentions the full name. You may want to mention it in line 51

AC: It has been done.

RC: L67 3096 X 2080 -> 3,096 X 2,080. You may need thousand separators.

AC: It has been corrected.

RC: L77 a space before “To”?

AC: A space has been added.

RC: L157 You may want no indent before “where”

AC: It has been corrected.

RC: L183 10,201thousand separators

RC: L184 10,201thousand separators

RC: L190 10,201thousand separators

AC: All thousands separators have been included.

RC: L338 difference -> different?

AC: Done.

---

## [Decision Letter · Decision Letter 1]

11 Mar 2022

ORION software tool for the geometrical calibration of all-sky cameras

PONE-D-21-34079R1

Dear Dr. Antuña-Sánchez,

We’re pleased to inform you that your manuscript has been judged scientifically suitable for publication and will be formally accepted for publication once it meets all outstanding technical requirements.

Kind regards,

Sergio Consoli

Academic Editor

PLOS ONE

Additional Editor Comments (optional):

Reviewers' comments:

Reviewer's Responses to Questions

**Comments to the Author**

1. If the authors have adequately addressed your comments raised in a previous round of review and you feel that this manuscript is now acceptable for publication, you may indicate that here to bypass the “Comments to the Author” section, enter your conflict of interest statement in the “Confidential to Editor” section, and submit your "Accept" recommendation.

Reviewer #1: All comments have been addressed

Reviewer #3: All comments have been addressed

2. Is the manuscript technically sound, and do the data support the conclusions?

Reviewer #1: Yes

Reviewer #3: Yes

3. Has the statistical analysis been performed appropriately and rigorously? 

Reviewer #1: Yes

Reviewer #3: Yes

4. Have the authors made all data underlying the findings in their manuscript fully available?

Reviewer #1: Yes

Reviewer #3: Yes

5. Is the manuscript presented in an intelligible fashion and written in standard English?

Reviewer #1: Yes

Reviewer #3: Yes

6. Review Comments to the Author

Reviewer #1: The manuscript is acceptable for publication now. The author has done a novel work in their field and PLOS ONE is a suitable jpurnal

Reviewer #3: This paper describes ORION, an open source software package created to calibrate all-sky cameras. ORION uses a set of photographs of the nighttime sky along with the known positions of stars to determine a mapping between image pixels and sky coordinates. ORION is meant to be low-cost, accurate, and easy to use. After a discussion of the theory behind geometric calibration, the paper presents an example of using ORION to generate the calibration matrices from a specific set of photographs.

L15: A definition/description of "hot pixels" would be useful to the reader.

L17: "kind of instrument", not "kind of instruments"

L31: "same size as the camera images", not "same size than the camera images"

L99-100: "on an image-by-image basis," not "on an image-by-image,"

L108: Add a space between the degrees symbol and the word "in". Missing spaces after a degree symbol also occurs on L135 (twice), L137, and L280 (twice).

L309-315: How do ORION's results compare to current geometrical calibration systems (mentioned on L38)?

7. PLOS authors have the option to publish the peer review history of their article (what does this mean?). If published, this will include your full peer review and any attached files.

Reviewer #1: No

Reviewer #3: No

---

## [Editor Report · Acceptance letter]

23 Mar 2022

PONE-D-21-34079R1 

ORION software tool for the geometrical calibration of all-sky cameras 

Dear Dr. Antuña-Sánchez:

I'm pleased to inform you that your manuscript has been deemed suitable for publication in PLOS ONE. Congratulations! Your manuscript is now with our production department. 

Kind regards, 

on behalf of

Dr. Sergio Consoli 

Academic Editor

PLOS ONE